# Detection of Bombali Virus in a *Mops condylurus* Bat in Kyela, Tanzania

**DOI:** 10.3390/v16081227

**Published:** 2024-07-31

**Authors:** Ariane Düx, Sudi E. Lwitiho, Ahidjo Ayouba, Caroline Röthemeier, Kevin Merkel, Sabrina Weiss, Guillaume Thaurignac, Angelika Lander, Leonce Kouadio, Kathrin Nowak, Victor Corman, Christian Drosten, Emmanuel Couacy-Hymann, Detlev H. Krüger, Andreas Kurth, Sébastien Calvignac-Spencer, Martine Peeters, Nyanda E. Ntinginya, Fabian H. Leendertz, Chacha Mangu

**Affiliations:** 1Helmholtz Institute for One Health, Helmholtz Center for Infection Research, 17489 Greifswald, Germany; kathrin.nowak@helmholtz-hioh.de (K.N.); sebastien.calvignac-spencer@helmholtz-hioh.de (S.C.-S.); fabian.leendertz@helmholtz-hioh.de (F.H.L.); 2Robert Koch Institute, 13353 Berlin, Germany; caroline.roethemeier@mdc-berlin.de (C.R.); merkelk@rki.de (K.M.); weisss@rki.de (S.W.); landera@rki.de (A.L.); kurtha@rki.de (A.K.); 3NIMR-Mbeya Medical Research Center, Mbeya P.O. Box 2410, Tanzania; lsudi@nimr-mmrc.org (S.E.L.); nelias@nimr-mmrc.org (N.E.N.); cmangu@nimr-mmrc.org (C.M.); 4TransVIHMI, Montpellier University/IRD/INSERM, 34394 Montpellier, France; ahidjo.ayouba@ird.fr (A.A.); guillaume.thaurignac@inserm.fr (G.T.); martine.peeters@ird.fr (M.P.); 5Max Delbrück Center for Molecular Medicine, 10115 Berlin, Germany; 6Institute of Virology, Charité—University Medicine Berlin, Corporate Member of Free University Berlin, Humboldt-University Berlin, 10117 Berlin, Germany; victor.corman@charite.de (V.C.); christian.drosten@charite.de (C.D.); detlev.krueger@charite.de (D.H.K.); 7Une Santé Pour Tous, Bingerville, Côte d’Ivoire; kleonce08@yahoo.fr (L.K.); chymann@gmail.com (E.C.-H.); 8Departement de Biologie Animale, Faculté des Sciences Biologiques, Université Peleforo Gon Coulibaly, Korhogo BP 1328, Côte d’Ivoire; 9Faculty of Mathematics and Natural Sciences, University of Greifswald, 17489 Greifswald, Germany

**Keywords:** *Orthoebolavirus bombaliense*, ebolavirus, *Filoviridae*, natural host, free-tailed bat

## Abstract

Bombali virus (BOMV) is a novel Orthoebolavirus that has been detected in free-tailed bats in Sierra Leone, Guinea, Kenya, and Mozambique. We screened our collection of 349 free-tailed bat lungs collected in Côte d’Ivoire and Tanzania for BOMV RNA and tested 228 bat blood samples for BOMV antibodies. We did not detect BOMV-specific antibodies but found BOMV RNA in a *Mops condylurus* bat from Tanzania, marking the first detection of an ebolavirus in this country. Our findings further expand the geographic range of BOMV and support *M. condylurus*’ role as a natural BOMV host.

## 1. Introduction

Ebola virus disease (EVD) is a severe disease with high case-fatality rates (25–90%) that sporadically occurs in Sub-Saharan Africa [1]. EVD outbreaks are caused by zoonotic spillovers of ebolaviruses from unknown animal reservoirs, and exceptionally, by resurgence from human EVD survivors [2,3]. Different bat species have been implicated as ebolavirus reservoir hosts by molecular, serological, and circumstantial findings, but conclusive evidence remains elusive [4,5,6].

In 2018, Bombali virus (BOMV), a novel Orthoebolavirus, was discovered in two free-tailed bat species (*Chaerephon pumilus* and *Mops condylurus*) from Bombali District, Sierra Leone [7]. Since then, the virus has also been detected in *M. condylurus* from Guinea, Kenya, and Mozambique [8,9,10]. A follow-up study in Kenya showed that BOMV was present in bats caught at the initial sampling site after one year [11]. Histopathological examination revealed no differences between PCR-positive and PCR-negative lung tissue, suggesting that BOMV infections in bats are mild [12]. The repeated and geographically widespread detections of asymptomatic BOMV infections point to *M. condylurus* and potentially other free-tailed bats as natural BOMV hosts.

These findings prompted us to screen our collection of free-tailed bat specimens sampled in Tanzania and Côte d’Ivoire between 2013 and 2017 for BOMV RNA and antibodies. Here, we report the detection of BOMV RNA in an *M. condylurus* bat from southern Tanzania. This finding extends the geographic range of BOMV and lends further support to the role of *M. condylurus* as its natural host.

## 2. Materials and Methods

### 2.1. Bat Sampling

Our collection comprised 349 free-tailed bats sampled for different studies in Tanzania and Côte d’Ivoire between 2013 and 2017 (Figure 1 and Appendix A).

Bats were caught with 12 m mist nets or self-made funnel traps. The captured bats were anesthetized using 10 mg/kg ketamine and 2 mg/kg xylazine and euthanized by exsanguination by cardiac puncture. Blood and organ samples were collected following strict biosafety protocols, snap-frozen in liquid nitrogen, and placed at −80 °C for long-term storage. All steps were performed by trained personnel under veterinary supervision and with permission by the relevant local authorities. In Tanzania, the study was conducted under the Clearance Certificate for Conducting Medical Research in Tanzania NIMR/HQ/R.8a/Vol. IX/2468. In Côte d’Ivoire, the ethics approval for sampling and analyzing small mammal organs was obtained from the National Commission of Ethics (CNER) under the number 033/MSLS/CNER-dkm.

### 2.2. Taxonomic Assignment

In the field, all bats were identified at least to the family level. To confirm and refine the taxonomic assignment, we performed a PCR targeting an ~800 bp fragment of the cytochrome B (CytB) gene (Table A1) and sequenced the PCR products using Sanger’s method. The resulting sequences were matched against the nucleotide collection of the National Center for Biotechnology Information (NCBI) using the basic local alignment search tool (BLASTn) [13].

Because not all sequences could be assigned to the species level using BLASTn, we performed a species delimitation analysis using a Bayesian Poisson tree process (bPTP) model [14]. Briefly, we selected CytB sequences generated in this study with a length ≥ 700 bp and ran cdhit to remove identical sequences [15]. We included *Myotis daubentoni* (AB106589) as an outgroup and aligned the CytB sequences with MAFFT implemented in Geneious Prime 2020.2.3 (https://www.geneious.com, accessed on 31 May 2021). The resulting alignment was used to build a maximum likelihood (ML) phylogenetic tree using IQ-TREE version 2.1.2 with automatic model selection [16]. TPM2 + F + G4 was selected as the best-fitting model. Branch robustness was assessed with Shimodaira–Hasegawa-like approximate likelihood ratio tests (SH-like aLRTs) with 100,000 replicates [17]. The ML tree was uploaded to the bPTP web server (https://species.h-its.org/, accessed on 31 May 2021) for species delimitation analyses [14]. We ran the bPTP model with 500,000 Markov chain Monte Carlo (MCMC) generations and checked for convergence.

### 2.3. Molecular Screening

For BOMV screening, we selected lung tissue based on previous studies indicating a lung tropism for BOMV and other viruses from the *Filoviridae* family in bats [8,9,11,18]. Extraction and cDNA synthesis were performed in two different laboratories, i.e., at the Robert Koch Institute, Berlin, and at the Institute of Virology of the Charité, Berlin. Therefore, different extraction and cDNA synthesis kits were used. At the Robert Koch Institute, we extracted full nucleic acids from lung tissue using the QIAamp^®^ Viral RNA Kit and generated cDNA using SuperScript IV Reverse Transcriptase (Invitrogen, Waltham, MA, USA) and random hexamer primers (Roche, Basel, Switzerland). At the Charité, we extracted RNA using the RNeasy Mini Kit (Qiagen, Hilden, Germany) and converted it to cDNA using M-MLV Reverse Transcriptase (Invitrogen) with random hexamer primers (Roche).

Samples were tested in duplicate with a BOMV-specific qPCR targeting a short region of the L gene (Table A1) [7]. Briefly, 5 µL of cDNA was added to a 20 µL master mix containing 11.65 µL of molecular-grade H_2_O, 2.5 µL of 10X Rxn buffer, 2 µL of MgCl_2_, 2 µL of dNuTPs (2.5 mM), 0.75 µL of each primer Filo_UCD_qFor (10 µM) and Filo_UCD_qRev (10 µM), 0.25 µL of the Filo_UCD_probe (10 µM), and 0.1 µL of Platinum™ Taq DNA Polymerase (Invitrogen). The cycling conditions were the following: 10 min initial denaturation at 95 °C followed by 45 cycles of 15 s at 95 °C and 60 s at 60 °C. As a standard for quantification, we included different dilutions of a synthetic DNA positive control (10^1^–10^6^ copies).

We used an in-house semi-nested PCR assay targeting a separate region of the L gene of members of the *Filoviridae* family to confirm a positive result from the BOMV-specific qPCR (Table A1). The semi-nested PCR assay was designed at the Institute of Virology of the Charité based on the RefSeq genomes of Tai Forest ebolavirus (NC_014372), BDBV (NC_014373), EBOV (NC_002549), SUDV (NC_006432), RESTV (NC_004161), the bat-infecting Lloviu cuevavirus (NC_016144), Marburg marburgvirus isolate Musoke (MARV; NC_001608), and Marburg marburgvirus isolate Ravn (NC_024781), as well as a bat-derived filoviral sequence (KP233864). Briefly, for the first round of PCR, we added 5 µL of extracted RNA to a 20 µL master mix containing 1.1 µL of molecular-grade H_2_O, 12.5 µL of 2× Reaction Mix, 0.4 µL of MgSO4 (50 mM), 1µL of Bovine serum albumin, 2 µL of each of primers PanFiloVMC_F2 (10 μM) and PanFiloVMC_R1(10 μM), and 1 µL of SuperScript™ III One-Step RT-PCR System with Platinum™ Taq DNA Polymerase (Invitrogen). After an initial reverse transcription for 20 min at 48 °C, we applied a touch-down approach with 50 cycles of 15 s denaturation at 98 °C, 20 s annealing starting at 60 °C and decreasing by 1 °C per cycle for the first 10 cycles, and 45 s elongation at 72 °C, followed by a final elongation for 2 min at 72 °C. We used 1 µL of the first-round PCR product in a 24 µL master mix consisting of 18.15 µL of H_2_O, 2.5 µL of 10× PCR buffer, 0.5 µL of dNTP mix (10 mM each), 1.25 µL of MgCl_2_ (50 mM), 0.75 µL of each of primers PanFiloVMC_F3 (10 μM) and PanFiloVMC_R1(10 μM), and 0.1 µL of Platinum™ Taq DNA Polymerase (Invitrogen). The cycling protocol was identical to the first-round PCR without the reverse transcription step. The sensitivity of the assay was assessed by serial dilution of two photometrically quantified in vitro transcribed RNAs of EBOV and MARV. The detection limit of the assay was 11.5 RNA copies/µL and 31.0 copies/µL, respectively. PCR products from the BOMV-specific qPCR and the semi-nested *Filoviridae* PCR were sequenced using Sanger’s method.

### 2.4. High-Throughput Sequencing

To generate additional genetic data, we sequenced the positive specimen on Illumina platforms with and without prior enrichment by in-solution hybridization capture.

For shotgun sequencing, we prepared a library using the KAPA RNA Hyper Prep Kit (Illumina, San Diego, CA, USA) according to the manufacturer’s instructions. Up to 100 ng RNAs were fragmented for 6 min at 85 °C and amplified for eight PCR cycles. The resulting libraries were quantified using the Qubit dsDNA HS Kit (Thermo Fisher Scientific, Waltham, MA, USA) and visualized using the High-Sensitivity D1000 Kit on an Agilent 4200 TapeStation System (Agilent Technologies, Santa Clara, CA, USA). The library was paired-end sequenced on an Illumina NextSeq with 150 cycles.

For target enrichment, we first performed DNase treatment on 30 µL of nucleic acid extract using the TURBO DNA-free™ Kit (Invitrogen), followed by clean-up with the RNA Clean and Concentrator-5 Kit (Zymo Research, Irvine, CA, USA), cDNA synthesis using the SuperScript™ IV First-Strand Synthesis System (Invitrogen) and second-strand synthesis with the NEBNext^®^ Ultra™ II Non-Directional RNA Second-Strand Synthesis Module (New England Biolabs, Ipswich, MA, USA). We fragmented 47.6 ng double-stranded cDNA with a Covaris S220 Focused ultrasonicator to generate 400 bp fragments and built a dual-indexed Illumina library using NEBNext^®^ Ultra™ II DNA Library Prep Kit from Illumina^®^ (New England Biolabs). The library was quantified using the KAPA Library Quantification Kit (Roche). Briefly, the 125 ng library was used as input for in-solution hybridization capture using the myBaits Hybridization Capture Kit (Daicel Arbor Biosciences, Ann Arbor, MI, USA) with custom RNA baits designed to cover genomes of different *Filoviridae* (Table A2). We performed two rounds of hybridization capture at 65 °C for 24 h. After each round, the capture product was amplified using the KAPA HiFi HotStart ReadyMix (Roche) with primers targeting the Illumina adapters and quantified using the KAPA Library Quantification Kit (Roche). The final product was sequenced on an Illumina MiniSeq platform using the MiniSeq High Output Reagent Kit for 300 cycles (Illumina).

The resulting sequencing reads were filtered using Trimmomatic [19] (settings: LEADING:30 TRAILING:30 SLIDINGWINDOW:4:30 MINLEN:40), merged with ClipAndMerge [20], and mapped to the Kenyan BOMV strain B241 (MK340750) using BWA-MEM [21].

### 2.5. Phylogenetic Analyses

To assess the relationship between the generated sequences and known BOMV strains, we performed maximum likelihood and Bayesian phylogenetic analysis. We assembled four different datasets for phylogenetic analysis: For ingroup analysis, we used all published complete and partial BOMV sequences together with all Sanger and Illumina sequences generated in this study (set1), or with only Sanger sequences (set2). For outgroup analysis, we added the reference genome of EBOV (NC_002549.1) to both datasets and only used coding sequences to allow for correct alignment (set3 and set4).

Sequences were aligned with MAFFT implemented in Geneious Prime. The resulting alignments were used to build maximum likelihood (ML) phylogenetic trees using IQ-TREE version 2.1.2 with automatic model selection [16]. Branch robustness was assessed with Shimodaira–Hasegawa-like approximate likelihood ratio tests (SH-like aLRTs) with 100,000 replicates [17]. For ingroup analyses (set1 and set2), TIM + F was selected as the best-fitting model.

To assess the placement of the root, we performed outgroup analyses (set3 and set4), with GTR + F + I as the best-fitting model. To confirm the tree topology, we performed Bayesian phylogenetic analyses on set1 and set2 in BEAST v1.10.4. [22]. For this, we built a model using BEAUti v1.10.4 assuming a strict molecular clock, constant population size with a lognormal distribution, and GTR with empirical base frequencies as the substitution model. We assessed the temporal signal in set1 and set2 using the Bayesian evaluation of temporal signal (BETS) test in BEAST with a generalized stepping-stone marginal likelihood estimation using 50 stepping stones in 50,000 generations [23]. A (log) Bayes factor of 1 for both set1 and set2 (Table A3) suggested that sampling dates covered an insufficient time span for crossing the phylodynamic threshold [24], and therefore tip dates were not used for calibration. Two independent MCMCs were run for 10,000,000 states, and the MCMC trace files were checked in Tracer v1.7.2 (24). Log and tree files were combined in LogCombiner v1.10.4, and TreeAnnotator 1.10.4 was used to summarize the posterior distributions of trees as maximum clade credibility (MCC) trees [22].

### 2.6. Virus Isolation

We attempted virus isolation in cell culture from lung, kidney, and blood samples of the BOMV-positive bat (performed under BSL-4 conditions). Vero cells and different *M. condylurus* cells, i.e., immortalized kidney cells (MoKi-Immortalized), primary lung cells (MoLu Prim), and primary brain cells (MoBra Prim) [25], were inoculated with tissue homogenate and checked for cytopathic effect every 48 h over ten days before being passaged (total of two passages). After each passage, a sample was taken and tested by BOMV-specific qPCR [7].

### 2.7. Serological Investigation

For antibody detection, 228 whole-blood samples of PCR-tested bats (Appendix A) were screened with a bat-adapted Luminex-based serological assay comprising recombinant antigens of five ebolaviruses (Ebola virus (EBOV), Sudan virus (SUDV), Bundibugyo virus (BDBV), Reston virus (RESTV), and BOMV) [26,27]. The following recombinant antigens were included in the assay: EBOV: glycoprotein (GP) of two strains (Mayinga from the Democratic Republic of the Congo, 1976, and Kissidougou/Makona from Guinea, 2014), 40 kDa protein (VP40), and nucleoprotein (NP); SUDV: GP, VP40, and NP; BDBV: GP and VP40; RESTV: GP; BOMV: GP. Blood samples were heat-inactivated at 56 °C for 30 min and diluted 1:2000 in assay buffer (phosphate-buffered saline (PBS) containing 0.75 mol/L of NaCl, 1% (*w*/*v*) bovine serum albumin (Sigma-Aldrich, St. Quentin Fallavier, France), 5% (*v*/*v*) fetal bovine serum (Gibco-Invitrogen, Cergy Pontoise, France), and 0.2% (*v*/*v*) Tween 20 (Sigma-Aldrich)). We mixed 100 μL of this dilution with 50 µL of recombinant protein-coated beads (2 µg protein/1.25 × 106 beads) and incubated for 16 h at 4 °C. After washing, we added 0.1 μg/mL of goat anti-bat biotin-labeled IgG (Euromedex, Souffelweyersheim, France) and incubated for 30 min. Following another washing step, we added 50 µL of 4 µg/mL streptavidin-R-phycoerythrin (Fisher Scientific) and incubated for 10 min. Fluorescence intensity was measured with BioPlex-200 (BioRad, Hercules, CA, USA) and expressed as median fluorescence intensity (MFI) per 100 beads [27]. In the absence of bat-positive controls, the cutoff median fluorescence intensity (MFI) was calculated from a set of 6000 bat blood samples for BOMV and more than 8000 bat blood samples for EBOV, SUDV, BDBV, and RESTV using (a) the mean MFI of negative control samples plus 4 times the standard deviation, (b) change point analyses, and (c,d) fitting of univariate distributions (negative binomial and negative exponential distribution) to our data; we defined the cutoff as a 0.001 risk for error, as previously described (14, 26). We defined the mean of the four methods as the cutoff for reactivity. Bats were considered seropositive for an ebolavirus only when reactive to glycoprotein (GP) and one additional protein [27], with the exception of RESTV and BOMV, for which only GP was available in the assay.

## 3. Results

### 3.1. Taxonomic Assignment

BLASTn analyses of the CytB sequences clearly identified 83 bats as belonging to the species *M. condylurus.* The other sequences either produced similarly good hits to several species within the genus *Chaerephon* (n = 157) or did not yield any highly similar hits (n = 109) (Table 1 and Appendix A).

The bPTP species delimitation analysis estimated four to nine species in our dataset (mean = 4.24), of which three species—including the outgroup—were supported by high posterior probability values (>0.95). The two well-supported free-tailed bat species covered the *M. condylurus* sequences (species 1) and the cluster of sequences without highly similar BLASTn hits (species 2) (Figure A1). Based on morphological traits and geographical distribution, species 2 was assigned as *Chaerephon* cf. *major*. A third cluster containing sequences highly similar to *Chaerephon* sp. received a slightly lower posterior probability of 0.85 (species 3) (Figure A1). According to the closest BLASTn hits, species 3 was assigned as *Chaerephon pumilus/leucogaster* group. Based on their very low posterior probability values (<0.13), we considered the remaining putative species as unlikely. The individuals excluded from phylogenetic and species delimitation analyses (i.e., <700 bp) could all be assigned to species 1–3 based on their sequence with the exception of CIV919, which could be assigned to the genus *Chaerephon* but did not cluster with the *C. pumilus*/*leucogaster* group (Table 1).

### 3.2. Molecular BOMV Testing and Sequencing

Using a BOMV-specific qPCR [7], 1/349 lung extracts were repeatedly weakly positive in one replicate (CT = 36.74 and 40.27, <10 copies/µL) (Table 1). The same extract was also positive in an in-house semi-nested PCR assay targeting the L gene of different filoviruses.

The positive sample originated from an *M. condylurus* bat (TZ154) from Kyela District in southern Tanzania. All other available sample types for TZ154 (i.e., liver, spleen, kidney, intestines, blood, and throat swab) were negative in the BOMV qPCR assay.

The Sanger sequencing of the PCR products produced two fragments of the L gene: a 106 nucleotides (nt) fragment for the BOMV qPCR (accession: PP175521), which differed from the positive control in two sites, and a 385 nt fragment for the semi-nested PCR (accession: PP175522) (Figure A2). High-throughput sequencing without prior enrichment (i.e., shotgun sequencing) did not produce any reads that could be mapped to the BOMV reference. After enrichment by hybridization capture, 70,059 (2.57%) reads were mapped to the reference, resulting in 19 unique reads after the removal of duplicate reads. This produced three additional short BOMV fragments of 102 nt (NP gene), 148 nt (L gene), and 154 nt (L gene) (accession: SUB14147099) albeit at low sequencing depth (3×) (Table 2 and Figure A2). We were unable to generate a complete BOMV genome, likely because of the very low viral copy numbers and the poor quality of the tissue, which had already been thawed twice before BOMV testing.

### 3.3. Phylogenetic Analyses

In all analyses, TZ154 formed a well-supported cluster with the Kenyan BOMV strains (Figure 2, Figure A3, Figure A4 and Figure A5). Outgroup analyses placed the root between the West African and East African BOMV strains but with low statistical support (Figure A4). The Bayesian phylogenetic tree confirmed this tree topology and separated BOMV strains in a West African and an East African clade (again with low statistical support) (Figure 2 and Figure A5).

### 3.4. Virus Isolation and Serology

No BOMV replication was detected in any of the used cell types and passages. The lack of success in virus culture attempts can likely be attributed to the low viral copy numbers and the poor quality of the tissue, which had already been thawed twice before BOMV testing.

While 3/228 bat blood samples were reactive to one or two ebolavirus proteins, none reacted to multiple proteins of the same ebolavirus and none to BOMV GP, including TZ154 (Table 1 and Appendix A).

## 4. Discussion

In this study, we tested samples of 349 free-tailed bats captured in Côte d’Ivoire and Tanzania for BOMV, including 83 *M. condylurus* bats. We screened lung samples of all 349 animals for BOMV RNA and tested 228 blood samples for antibodies against various ebolaviruses. We detected BOMV-RNA in a lung sample of an *M. condylurus* bat captured in Tanzania in 2017. The short sequences (in total 895 nt) generated by Sanger and Illumina sequencing formed a well-supported clade with BOMV strains from Kenya.

The very low BOMV copy numbers detected in TZ154, along with the limited success of high-throughput sequencing efforts to recover BOMV reads and the failure of virus culture attempts, may simply be a result of the poor tissue quality. Indeed, the positive lung sample had already been thawed twice before BOMV testing. Nonetheless, the detection of BOMV RNA using three different methods (i.e., two distinct PCR systems and high-throughput sequencing) in two laboratories, which have never previously handled BOMV-positive specimens, lends credibility to our findings.

We did not detect BOMV-specific antibodies in this study. This does not necessarily imply that BOMV infections are uncommon in free-tailed bats but could also be attributed to the small sample size, especially when considering only *M. condylurus* (n = 83). Additionally, for the Marburg virus, another virus in the *Filoviridae* family, detectable antibody levels in its fruit bat reservoir, *Rousettus aegyptiacus*, wane within three months [28]. Though no data are available, this rapid waning of detectable immunity could theoretically also be the case for the related BOMV. Finally, BOMV has not been cultured to date, and no positive controls were available, which means that the recombinant BOMV GP could not be validated under the same conditions as the other ebolavirus proteins. Thus, a technical problem, though unlikely, cannot be excluded. The fact that the PCR-positive bat was seronegative for BOMV GP suggests that we caught the animal in the early stages of infection before it developed detectable IgG antibodies (no IgM assay was available). This could also explain the low BOMV copy numbers detected in this animal (TZ154).

Interestingly, TZ154 showed reactivity to EBOV NP. This may be due to cross-reactivity to BOMV, for which a specific NP was not available (Appendix A, Figure A6). In addition, two *Chaerephon* sp. bats from Côte d’Ivoire were reactive to SUDV NP (CIV903) and to both EBOV GP and RESTV GP (CIV1192). This may indicate exposure to SUDV, EBOV, or other known or unknown ebolaviruses. Exposure to RESTV is extremely unlikely as RESTV is not endemic in Africa. Overall, serological cross-reactivity between different ebolaviruses is common [27,29], and unspecific reactivity may also occur, as suggested by the finding of EBOV seroreactivity among 2% of dogs tested in France [30]. Thus, we suggest interpreting our serological findings with some caution.

The detection of BOMV RNA in an *M. condylurus* bat in Kyela District, Tanzania, suggests that BOMV is widespread throughout the species’ range and adds to the growing body of evidence indicating *M. condylurus* as a natural BOMV host. Kyela District is located approximately 850 km southwest of the closest BOMV detection site in Taita Hills, Kenya, and approximately 1260 km north of the southernmost detection in the Inhassoro District, Mozambique.

Lebarbenchon et al. hypothesized a potential seasonality of BOMV based on the observation that all BOMV-positive bats were caught in May [10], which was also the case for the positive bat in this study. While at least in our case this may reflect a seasonal sampling bias (54/70 *M. condylurus* from Tanzania were caught in May), it remains an intriguing hypothesis for future studies to explore.

BOMV is the first ebolavirus reported in Tanzania. While it is unknown if BOMV can infect humans, the peri-urban lifestyle of *M. condylurus* certainly provides opportunities for spillover [7,31]. Indeed, TZ154 was caught from a large palm tree harboring an *M. condylurus* colony with several hundred individuals located close to a small neighborhood where people reported the use of guano collected under the tree as fertilizer. Overall, Kyela District is relatively densely populated, and *M. condylurus* and other free-tailed bats are abundant in houses, churches, warehouses, and other public buildings.

The repeated findings of BOMV in *M. condylurus* across Africa clearly link an ebolavirus to a specific natural host. Future investigations of BOMV in *M. condylurus* and the human population in affected areas offer a unique opportunity to address long-standing questions on ebolavirus ecology in their hosts and can help to elucidate factors that facilitate spillover events.

## Figures and Tables

**Figure 1 viruses-16-01227-f001:**
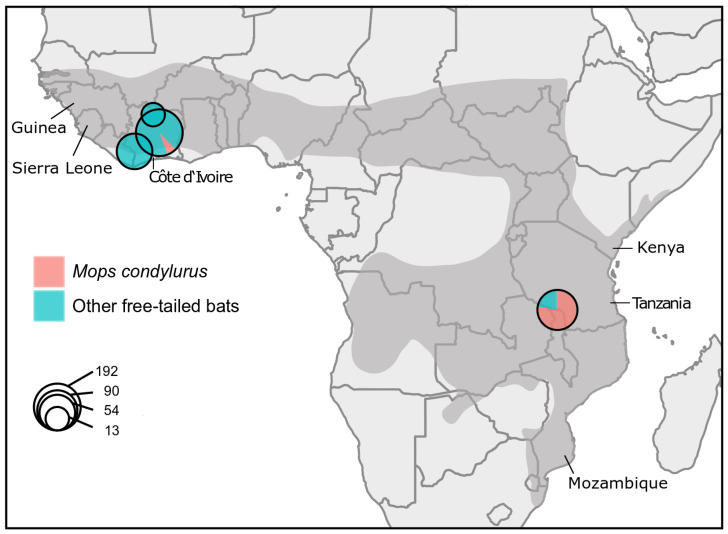
Map of bat sampling sites. The pie chart size corresponds to the number of individuals sampled at each site (log scale); *M. condylurus* in red and other free-tailed bats in blue. *M. condylurus* distribution is marked in dark gray.

**Figure 2 viruses-16-01227-f002:**
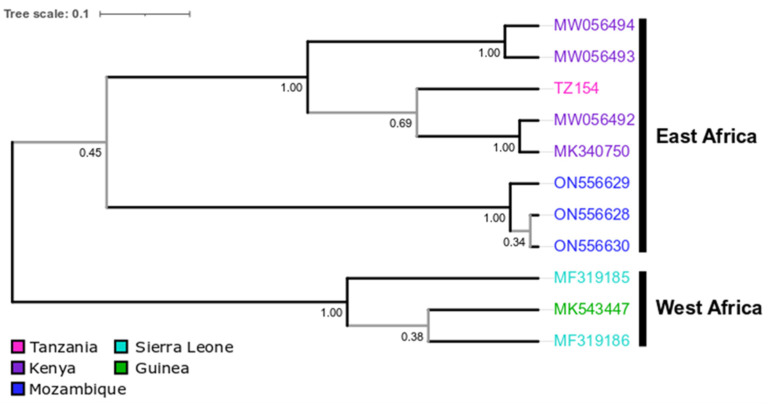
Bayesian phylogenetic tree of BOMV sequences (set1). Maximum clade credibility tree summarized from BEAST analysis using all published partial and complete BOMV sequences and all sequences generated from TZ154. Leaves are colored according to country and labeled with strain name or accession number. Posterior probability (PP) is given beside nodes (PP < 0.95 is marked in gray). Scale bar indicates substitutions per site.

**Table 1 viruses-16-01227-t001:** Overview of sampling information and laboratory results. Positive test results are marked in bold.

Species	Country	BOMV PCR (n Positive/n Tested)	Seroreactivity (Antigen)
*M. condylurus*	Tanzania	**1/70**	**1/53** (EBOV NP)
	Côte d’Ivoire	0/13	0/12
*Chaerephon pumilus*/*leucogaster* group	Tanzania	0/20	0/14
	Côte d’Ivoire	0/89	**1/33** (SUDV NP)
*Chaerephon* cf. *major*	Tanzania	0/0	0/0
	Côte d’Ivoire	0/156	**1/115** (EBOV GP; RESTV GP)
*Chaerephon* sp.	Tanzania	0/0	0/0
	Côte d’Ivoire	0/1	0/1

**Table 2 viruses-16-01227-t002:** High-throughput sequencing results on Illumina platforms.

	Shotgun Sequencing	Target Enrichment
Total reads after trimming	25,047,869	2,725,836
Reads mapped to BOMV reference genome	0	70,059
Unique reads mapped to BOMV reference genome	0	19
Sites covered 1×	0	411
Sites covered 3×	0	404

## Data Availability

Sequence data are available under the NCBI accession numbers PP175521, PP175522, and SUB14147099.

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
