# Peer review of "Detection of Bombali Virus in a Mops condylurus Bat in Kyela, Tanzania"

_viruses, 2024, doi:10.3390/v16081227_

Round 1

Reviewer 1 Report

Comments and Suggestions for Authors

This study suggests Bombali virus infection in Mops Condylurus Bats in Tanzania. This study could be a useful insight into the distribution and natural hosts of Orthoebolaviruses. However, there are several concerns.

Major comments

1. From TZ154 sample, which region of the BOMV genome was detected? Mapped read count plot would be easy to understand.

2. What is the definition of “unique” in line 281. Please state clearly.

3. For the three antibody-positive animals, the authors should discuss more about what possible viral infections they may have had.

4. The authors should discuss the very low viral RNA level of TZ154. Is it due to the condition of the sample? Is it because TZ154 was at the early stage of BOMV infection as described lines 315-317? Or Is it because another animal is the natural host of BOMV?

Minor comments

5. Since readers may not be familiar with the geography of Africa, the authors should add the names of other countries where BOMV has been detected (i.e., Guinea, Kenya, and Mozambique).

6. Lines 307-317 should be moved to the Discussion section.

Author Response

Comment 1: From TZ154 sample, which region of the BOMV genome was detected? Mapped read count plot would be easy to understand.

Response 1: Thank you for this comment. To clarify which regions of the BOMV genome were detected, we adjusted the text and added a coverage plot as a figure to the Appendix A (Figure Appendix A2).

  • In the methods section on molecular testing, we added that the BOMV-specific qPCR targets a region of the L gene (lines 122-123), and that the semi-nested Filoviridae PCR targets a different region of the L gene (line 130).
  • In the results section on molecular BOMV testing and sequencing, we added that we produced two fragments of the L gene by Sanger sequencing of the PCR products (line 277), and 3 fragments – one of the NP gene and 2 of the L gene - by high throughput sequencing (line 284).

Comment 2: What is the definition of “unique” in line 281. Please state clearly.

Response 2: Unique reads refers to reads after the removal of identical duplicate reads. We added “after the removal of duplicate reads” to clarify the sentence (see line 283).

Comment 3: For the three antibody-positive animals, the authors should discuss more about what possible viral infections they may have had.

Response 3: Thank you for this comment. We agree that discussing seroreactivity improves the manuscript. We added information about the two additional seroreactive bats to the discussion, and discussed what may have led to these results, including infection with the respective ebolavirus, crossreactivity, and unspecific reactivity (lines 351 -358). We also included two references [29, 30] on cross-reactivity and unspecific reactivity.

Comment 4: The authors should discuss the very low viral RNA level of TZ154. Is it due to the condition of the sample? Is it because TZ154 was at the early stage of BOMV infection as described lines 315-317? Or Is it because another animal is the natural host of BOMV?

Response 4: Thank you for this comment. In the original version of the manuscript, we have mentioned the low quality of the tissue in the results section (3.2 and 3.4). We now added information about the poor tissues quality to the discussion, and highlight, why our findings are robust despite the low copy numbers, limited sequence data, and unsuccessful virus culture (lines 330-336). We also added a sentence that the low copy numbers may be due to the early stages of infection – as indicated by the lack of detectable BOMV IgG (lines 348/ 349). We did not discuss the low copy numbers in relation to the question if Mops condylurus is the natural host of BOMV or not, because a) other studies have detected high BOMV copy numbers in M. condylurus, and b) we have no knowledge how ebolaviruses behave in their natural hosts, and therefore cannot anticipate high or low copy numbers.

Comment 5: Since readers may not be familiar with the geography of Africa, the authors should add the names of other countries where BOMV has been detected (i.e., Guinea, Kenya, and Mozambique).

Response 5: In Figure 1, we added country names to all countries where BOMV has previously been detected (i.e. Guinea, Sierra Leone, Kenya, Mozambique).

Comment 6: Lines 307-317 should be moved to the Discussion section.

Comment 6: We moved lines 310-322 (former lines 307-317) to the Discussion section.

Reviewer 2 Report

Comments and Suggestions for Authors

The manuscript by Düx et al reports the identification of the Bombali virus in a histological sample taken from Mops condylurus. The epidemiological study of filoviruses in potential animal reservoirs is a topic of great interest, both to clarify the spread of these viruses and to consolidate knowledge on reservoirs.

The work is very interesting and the positive result expands knowledge on the spread of BOMV. Although the positive result was obtained in a single sample and in the presence of a very low quantity of viral RNA, the authors' efforts seem to confirm the result, thus providing new information on the spread of BOMV.

I suggest that the authors verify the use of italics for the species, also including the captions of the figures.

Author Response

Comment 1: I suggest that the authors verify the use of italics for the species, also including the captions of the figures.

Response 1: We would like to thank you for reviewing our manuscript and providing us with your feedback. As suggested, we checked the text for for the use of italics in species names. We identified the following seven species names that were not italicized, and corrected them:

  • Line 79: M. condylurus, M. condylurus
  • Line 101: Myotis daubentoni
  • Line 263: Chaerephon pumilus/ leucogaster
  • Line 267/ 268: C. pumilus/ leucogaster
  • Line 372: M. condylurus
  • Line 413: M. daubentoni

In addition, we adjusted the genus and species names in Table B2 to match the current ICTV taxonomy.